# Nutrient-Induced Cellular Mechanisms of Gut Hormone Secretion

**DOI:** 10.3390/nu13030883

**Published:** 2021-03-09

**Authors:** Van B. Lu, Fiona M. Gribble, Frank Reimann

**Affiliations:** Wellcome Trust-MRC Institute of Metabolic Science Metabolic Research Laboratories, Cambridge University, Addenbrookes Hospital, Cambridge CB2 0QQ, UK; vl285@medschl.cam.ac.uk

**Keywords:** enteroendocrine cells, chemosensory, GIP, GLP-1, nutrients, hormones

## Abstract

The gastrointestinal tract can assess the nutrient composition of ingested food. The nutrient-sensing mechanisms in specialised epithelial cells lining the gastrointestinal tract, the enteroendocrine cells, trigger the release of gut hormones that provide important local and central feedback signals to regulate nutrient utilisation and feeding behaviour. The evidence for nutrient-stimulated secretion of two of the most studied gut hormones, glucagon-like peptide 1 (GLP-1) and glucose-dependent insulinotropic polypeptide (GIP), along with the known cellular mechanisms in enteroendocrine cells recruited by nutrients, will be the focus of this review. The mechanisms involved range from electrogenic transporters, ion channel modulation and nutrient-activated G-protein coupled receptors that converge on the release machinery controlling hormone secretion. Elucidation of these mechanisms will provide much needed insight into postprandial physiology and identify tractable dietary approaches to potentially manage nutrition and satiety by altering the secreted gut hormone profile.

## 1. Introduction

The food we eat is composed of water and macronutrients including carbohydrates, fats, and proteins. These nutrients trigger physiological responses to initiate digestion, absorption, and metabolism of nutrients to allow for their biochemical utilisation in the body. Furthermore, nutrients activate neuronal and hormonal signalling to the brain to regulate food intake and appetite. The gastrointestinal tract plays a key role in mediating the physiological effects induced by ingested nutrients. It has long been long known that specialised cells lining the gut epithelium can sense changes in luminal content and respond by releasing chemicals. Bayliss and Starling [1] described the first gut hormone secretin, and demonstrated its release following delivery of acidic solutions into the small intestine. Similarly, nutrients ingested or liberated following digestion can stimulate hormone secretion from enteroendocrine cells (EECs), which are specialised gut epithelial cells that reside within the polarized absorptive epithelial layer. The anatomy of “open” type EECs, with a slender apical process that extends to the intestinal lumen and a basolateral surface facing the interstitial space and circulatory system, link luminal composition to a variety of secreted chemical signals thus making EECs prime candidates to serve as intestinal nutrient sensors. EECs may also form additional extensions that interact with local neuronal and glial cells [2,3] to further expand the range of physiological responses to detected nutrients. 

Gut responses triggered by nutrients extend beyond detection of the physical presence of substances within the intestinal lumen. Although classical studies demonstrated osmotic pressure within the stomach as an important factor in determining the rate of gastric emptying into the duodenum [4], the addition of physiological or hyperosmotic solutions of sodium chloride into the intestine were not sufficient to trigger gut hormone secretion [5], supporting the notion that nutrient-stimulated hormone release involves specific mechanisms. Moreover, nutrient-stimulated release can be disrupted using pharmacological and genetic approaches targeting transporter and carrier proteins as well as luminal and epithelial enzymes involved in digestion and absorption. This chapter will review the cellular mechanisms recruited by various nutrients to stimulate hormone secretion from the gastrointestinal tract, with a focus on mechanisms within EECs that release two important gut hormones implicated in glucose homeostasis and appetite regulation: glucagon-like peptide 1 (GLP-1) and glucose-dependent insulinotropic polypeptide (formerly known as gastric inhibitory peptide, GIP). 

## 2. Enteroendocrine Cells That Release GLP-1 and GIP

EECs are a collection of endocrine cells of the gastrointestinal tract, releasing over 30 different hormones into the bloodstream [6] that act locally, peripherally or centrally to initiate physiological responses. Although EECs are sparse and account for only 1% of the total epithelial cell number, they are scattered along the entire gastrointestinal tract from the stomach to the rectum covering a substantial area and thus together comprise the largest endocrine organ in the body. EECs arise from local intestinal stem cells and are in a continuous state of cell turnover, being replaced every 3–5 days in the small intestine [7]. 

Collectively, EECs secrete a range of hormones that regulate glucose homeostasis, gut motility, appetite, adiposity, and epithelial cell proliferation. Notably, GLP-1 and GIP act as incretin hormones, amplifying insulin secretion following oral glucose administration [8,9], and account for 50–70% of total postprandial insulin secretion. Exaggerated incretin hormone release following bariatric surgery contributes to the beneficial outcomes of weight loss [10] and glycaemic control [11] and mimetics of GLP-1 are effective treatments for Type 2 diabetes [12], with some agents additionally licenced to treat obesity. 

EECs are traditionally categorised into distinct cell types based on their hormonal signature. For instance, EECs that release GIP or GLP-1 are traditionally classified as K- and L-cells, respectively. However, immunohistochemical studies [13,14] and transcriptomic profiling of different EEC populations [15,16,17] have revealed an unexpected degree of overlap between EECs within the proximal small intestine, including those expressing GLP-1 and GIP. Indeed, the data suggest that individual EECs can express a much broader range of gut hormones than originally believed, which may be exploited in future therapeutic strategies. Although individual gut hormones are produced by overlapping populations of EECs, they each have a distinct longitudinal distribution along the gut [18,19,20] with the highest number of GIP-producing K-cells being found in the proximal small intestine, predominantly the duodenum [21,22], and GLP-1-producing L-cells more broadly located along the gut but increasing in numbers more distally with the highest density of L-cells in the distal small intestine and colon [23] (Figure 1). 

EECs in the proximal gut are well placed to respond acutely to incoming nutrient loads and are postulated to contribute more than distally located EECs to nutrient-driven satiety [24]. Most ingested nutrients are absorbed within the proximal small intestine and compared with the distal gut, the upper small intestine receives more vagal afferent innervation, which forms part of a neural circuit that mediates satiety [25,26]. The physiological roles of nutrient-sensing mechanisms in EECs of the distal gut remain elusive, although colonic EECs may respond to locally produced microbial products [27] and lipid metabolites [28] and provide signals reflective of long-term dietary history. However, the beneficial metabolic effects of increased nutrient exposure and recruitment of distal EECs following bariatric surgery suggest these mechanisms may be exploited for effective weight and blood glucose control.

This review will focus on mechanisms of GLP-1 and GIP secretion and suppression of appetite and food intake, but it is worth mentioning other co-released gut hormones from EECs that also mediate physiological effects on food intake and may act in concert with GIP and GLP-1. 

Cholecystokinin (CCK) is a peptide hormone secreted by a subset of EECs, classically designated as I-cells [29]; though significant protein levels of CCK are also produced in the brain and peripheral nervous system [30,31]. CCK plays an important role facilitating digestion in the small intestine by stimulating bile release from the gallbladder and enzyme secretion from the pancreas [32], and CCK also reduces food intake when administered peripherally [33,34] or centrally [35] via a number of mechanisms [36]. CCK is produced along the entire small intestine, with the highest density of CCK hormone-expressing cells in the duodenum [19,20,37]. Several transcriptomic [16,17,37] and immunohistochemical [37,38] studies demonstrated co-expression of CCK with other gut hormones including GIP, GLP-1, secretin and neurotensin. A recent transcriptomic analysis of all EEC populations in the small intestine revealed overlap of CCK expression in the majority of EEC cell types defined by hormonal expression profile [15]. The co-expression of CCK and incretin hormones within the same individual EECs suggests similar nutrient-sensing mechanisms can trigger multiple hormone release. However, carbohydrates were a modest secretagogue of CCK release despite being a potent stimulus for GIP and GLP-1 secretion [39,40,41]. Within EECs expressing multiple gut hormones, storage vesicles containing only CCK have been reported [42]; however, possible mechanisms to selectively mobilise specific vesicular pools remain to be elucidated. 

Secretin is another peptide hormone transcriptionally co-expressed with a variety of gut peptides [15], including GLP-1 and GIP [16,17,37,43], which switches on during EEC maturation [44]. Levels of secretin are elevated postprandially and play roles in gastric acid secretion, pancreatic bicarbonate release [45] and promoting satiety. Peripheral administration of secretin decreased food intake in rats [46,47] which was mediated by vagal afferent signalling [48]. Secretin producing EECs (S-cells) are found throughout the small intestine, with the highest immunoreactivity in the duodenum [19,20,37] and are also found in the colon of adult and developing mice [49]. 

Xenin [50] is a 25 amino acid length neurotensin-like peptide released from GIP-expressing K-cells [51], although of questionable physiological significance as it is produced from a cytoplasmic coat protein [52] with no clear evidence of how it might reach the lumen of secretory vesicles. Secretion of xenin is elevated after a meal and possibly triggered by the anticipation of food [53]. Xenin has been reported to enhance GIP-mediated insulin secretion [54] via activation of cholinergic neurons innervating β cells. Intravenous (IV) injection of synthetic xenin stimulated jejunal motility in dogs [55] and increased contraction frequency in humans [56]. Intracerebroventricular (ICV) injection of xenin reduced food intake and weight gain in mice and this effect was abolished in neurotensin receptor 1 (Ntsr1)-deficient mice [57,58]. Neurotensin (NTS) is a peptide hormone widely distributed in the central and peripheral nervous system and expressed in a subpopulation of EECs (N-cells). Both glucose and fat triggered NTS release [59,60,61] and centrally administered NTS reduces appetite [57,62,63], with peripheral satiety effects mediated primarily by Ntsr1 receptors located on vagal afferent neurons [58,64]. The greatest expression of NTS protein is found in the ileum [19,37,65] and co-localises with a number of gut hormones including the incretin hormones GLP-1 and GIP [16,17,37,66]. Interestingly, NTS was localised to a population of vesicles distinct from those staining for GLP-1 in the distal ileum [66], which may result from EECs expressing *Gcg* and *Nts* at different times during development and maturation. Although the independent release of NTS from this distinct pool of vesicles may be possible, GLP-1 and NTS were found to be co-secreted across a range of different stimuli [66]. 

Oxyntomodulin (OXM) is a circulating gut hormone produced from the same proglucagon precursor peptide as GLP-1, and can activate both GLP-1 and glucagon receptors (GLP1R, GCGR, respectively) [67]. ICV and intraperitoneal (IP) injections of oxyntomodulin in rats inhibit food intake and promote weight loss [68,69] and in mouse models, both GCGR and GLP1R activity were shown to contribute to the weight loss phenotype [70]. In a randomised double-blind placebo controlled cross-over study, IV oxyntomodulin administration reduced energy intake and significantly reduced hunger scores [71]. Long acting peptides combining GCGR and GLP1R activity are in clinical trials for the treatment of type 2 diabetes and obesity [72].

Peptide YY (PYY) hormone is co-located with GLP-1 in L-cells, and is found at highest levels in the ileum and colon [18,19,20] where it is co-released with GLP-1 [73,74,75]. IP or IV administration of PYY3-36 suppressed food intake in rodents and humans [76,77], analogous to the effects of GLP-1. Interestingly, direct stimulation of gut hormone release from distal L-cells, which increased both GLP-1 and PYY levels, reduced food intake as a result of PYY signalling through Y2 receptors [78]. However, not all studies have been able to reproduce the anorexigenic actions of exogenously administered PYY [79,80,81]. In studies describing a PYY-induced reduction in feeding, mechanisms proposed to mediate this effect include the vagal-brainstem-hypothalamic circuit [25,82,83], inhibition of gastric acid secretion [84,85] and delayed gastrointestinal motility/small intestinal transit [86,87,88]. 

Another hormonal product co-expressed in GLP-1 expressing L-cells of the distal colon and rectum is insulin-like peptide 5 (INSL5). INSL5 protein is co-stored in the same vesicles and co-released with GLP-1 and PYY following stimulation [74]. Unlike GLP-1 and PYY, however, INSL5 appears to exert orexigenic actions, as peripheral injection of INSL5 increased food intake in mice, an effect lost in mice deficient in its cognate receptor, relaxin family peptide receptor 4, Rxfp4 [89]. The overall importance of INSL5 in regulating feeding behaviour is unclear as ablation of *Insl5* expression in mice resulted in no obvious feeding deficit [90] and a subtle orexigenic effect of selective distal L-cell stimulation became apparent only when the opposing and overriding anorexigenic PYY effect was blocked with a Y2R-inhibitor [78]. The physiological rationale for distal EECs co-releasing hormones with opposing actions on food intake has yet to be reconciled. 

## 3. Models to Study Nutrient-Sensing Mechanisms in the Gut

A variety of in vitro and ex vivo experimental models have been developed to study the mechanisms regulating EECs function. In combination with in vivo models and clinical studies that measure food intake following ingestion of nutrients, a comprehensive understanding of nutrient-stimulated responses in the gastrointestinal tract has been revealed. 

Studies utilising intestinal cell line models of EECs provided initial insights into nutrient-sensing mechanisms in EECs. The most commonly used murine and human models for GLP-1 secretion are GLUTag [91] and NCI-H716 cells [92,93], respectively. GLUTag cells, derived from oncogenic tumours from the large bowel, respond to a range of nutrient and hormonal stimuli [94], whereas NCI-H716 cells, derived from a poorly differentiated adenocarcinoma of the human caecum, also respond to a range of nutrient stimuli [95] but possess altered regulation of proglucagon gene expression [96]. The secretin tumour cell line, STC-1 [97], derived from a mouse small intestinal neuroendocrine carcinoma, secretes a variety of small intestinal gut hormones including GIP, GLP-1, secretin and CCK. Subclones of STC-1 cells have been generated to produce lines with increased GIP expression or secretion [98,99,100]. Conflicting reports on the sensitivity of STC-1 cells to glucose emphasized the need for studies using primary intestinal cell models. Initial attempts to culture primary EECs involved elutriation to purify and enrich the K- or L-cell population from canine intestinal epithelia [101,102] or the use of fetal rat intestinal tissue [103]. Optimised culturing techniques, including enrichment of intestinal crypts or the deep folds of the intestinal epithelium have permitted studies from primary cultures of adult mouse [104] and human [105] intestinal tissue. The development of transgenic mouse models labelling K- and L-cell populations [104,106] has allowed transcriptomic and single-cell recording approaches to be applied to specific EEC populations and advanced our understanding of the signalling pathways recruited following specific nutrient exposure. Renewable cell culture technologies such as intestinal organoids [107] have been used to study EECs and have confirmed many nutrient-stimulated signalling mechanisms described in primary intestinal preparations [108] but more importantly provide a means to interrogate nutrient-sensing mechanisms in human EEC populations by labelling specific populations of EECs with fluorescent proteins under the control of hormonal promoters [109,110].

Two-dimensional cultures of these in vitro cell models permit access to EECs for membrane recordings and intracellular measurements. They also allow nutrients to bind targets on EECs that may be inaccessible from the luminal side of the intestine, by-passing nutrient absorptive and transport mechanisms. Three-dimensional organoid models maintain a polarized epithelium in culture, and as the luminal epithelial surface is directed towards the central organoid domain, exogenously applied nutrients may still by-pass transport and absorptive pathways. To investigate nutrient-sensing mechanisms under more physiological settings, ex vivo preparations such as Ussing chambers [111,112,113] and vascular perfused intestinal models have been used by several laboratories [114,115]. Properly prepared, both models retain tight junctions between epithelial cells and maintain the polarity of the epithelial layer. The location of receptors and sites of action of nutrients can be determined in these models by application of nutrients to the isolated apical or basolateral surface. Hormone secretion from L-cells has been monitored in these models either by collection of perfusate or media from the basolateral side combined with immunoassays [112], or by measurement of trans-epithelial short circuit currents [113].

Finally, in vivo models allow assessment of nutrient ingestion or infusion and effects on food intake and feeding behaviour. In rodents and humans, diets may be altered to control for macronutrient composition, but the palatability and other sensory attributes of food (e.g., odour) should be considered, as inputs from lingual sensors and olfactory centres can also impact feeding behaviour. Nutrients may also be delivered directly into the intestine by oral gavage or by insertion of a nasoenteral feeding tube into the stomach, duodenum, or jejunum. Alternatively, catheters may be surgically inserted to directly infuse nutrients into the lumen of a specified intestinal region, which however, if infused distal to the sphincter of Oddi, may not mix physiologically with pancreatic exocrine secretions and bile, thus altering the digestion of macronutrients. Studies involving human participants are relevant in translating our understanding of the cellular mechanisms underlying nutrient-sensing but can be particularly challenging. Measurements of hunger are subjective and dependent on previous eating habits. Moreover, multiple factors such as alterations in gut motility may unexpectedly modify gastrointestinal responses to nutrients. For instance, the effect of nutrients on hunger scores was attenuated in older participants [116].

## 4. Cellular Mechanisms of Nutrient-Induced Gut Hormone Secretion

The suppression of food intake following oral ingestion or direct infusion of nutrients into the intestine is associated with a rise in the release of GLP-1 and GIP [117,118,119]. Nutrient-induced gut hormone secretion entails recruitment of specific cellular mechanisms that are influenced by rates of digestion and absorption as well as expression of nutrient-specific transporters and receptors. 

### 4.1. Carbohydrates

Complex carbohydrates cannot be absorbed across the intestinal wall and therefore must be broken down to monosaccharides before they are transported across cell membranes. Amylases from saliva and pancreatic secretions initiate enzymatic digestion and a combination of hydrolases expressed on enterocytes complete the breakdown of ingested carbohydrates to monosaccharides.

Mechanisms of glucose-stimulated GLP-1 and GIP secretion have been intensively investigated as glucose is a potent secretagogue and there is interest in understanding the role of both incretin hormones in the pathology and therapeutics of Type 2 diabetes (T2D). In humans, glucose can be detected in the proximal duodenum within 5 min of ingestion of liquid glucose load, correlating with the time for first rapid elevation of GLP-1 and GIP concentrations in the bloodstream [120]. The early rise in GIP levels is readily attributed to arrival of nutrients in the duodenum with its high local density of GIP-expressing K-cells. However, the rapid rise in GLP-1 occurs well before glucose can reach the distal portion of the gut, and most of the glucose ingested is absorbed in the proximal small intestine with very little passing more distally to where the majority of GLP-1 expressing L-cells reside [120]. Measurements of glucose concentrations in the distal gut of several species after a meal confirm low levels of glucose [121]; however, levels at the ileo-caecal junction can rise as high as 10 mM after a meal [122]. The most likely mechanism to account for the rapid phase of GLP-1 secretion is the activation of proximal GLP-1 expressing cells [123], rather than recruitment of neuronal or humoral signals from the proximal to distal intestine, as patients with ileal resections maintained the rapid phase of GLP-1 secretion [124]. 

Glucose-stimulated release of GLP-1 and GIP from EECs requires absorption of carbohydrates. Monosaccharides are transported into cells by active and facilitative glucose transporters. Active glucose transporters, including the sodium-glucose linked transporter 1 (SGLT1), are located on the apical surface of the intestinal epithelial layer [125,126,127] and *Sglt1* is expressed in GIP- and GLP-1-releasing EECs [104,106,110] (Figure 2). Sodium-coupled active transporters carry glucose into cells, against its concentration gradient, along with sodium ions (Na^+^) down the Na^+^ concentration gradient established by basolateral Na/K-ATPase activity. The movement of 2 Na^+^ ions per glucose molecule by SGLT1 produces a small electrogenic current which is sufficient to depolarise the cell membrane potential and trigger action potential firing in GLUTag cells [128]. The sensitivity of L-cells to release GLP-1 in response to glucose closely mirrors the binding potency of SGLT1 for glucose [129] supporting the role of SGLT1 as the primary glucose-sensing mechanism for GLP-1 expressing cells. Non-metabolisable sugars such as alpha-methyl-glucopyranoside and 3-O-methylglucose [130,131] are also substrates for SGLT1 and stimulate GIP and GLP-1 secretion in vivo [115,132] and activating SGLT1 using various substrates reduced food intake [24]. Further supporting the essential role of SGLT1 in glucose-stimulated incretin hormone release, the SGLT1 inhibitor phloridzin or knock-out of *Sglt1* reduced glucose-triggered GLP-1 and GIP release [115,132,133]. However, whilst glucose triggered GIP-secretion is essentially absent in *Sglt1*-knock-out mice, the profile of GLP-1 release after an oral glucose challenge in *Sglt1*-deficient mice is complex as there is significantly reduced GLP-1 release at early time points (<15 min), but elevated GLP-1 release at later times (1–2 h) [134]. The amplified delayed phase of GLP-1 release in this model is thought to be due to reduced glucose absorption in the proximal gut and delivery of more glucose to the distal intestine that seems to activate SGLT1-independent glucose-sensing mechanisms, possibly involving fermentation to short chain fatty acids. Another sodium-glucose co-transporter, SGLT3, has been described in GLUTag cells [128] and enteric neurons of the submucosal and myenteric plexus [135], but whether SGLT3 contributes to glucose-sensing in EECs is not clear. Interestingly, the human variant of SGLT3 has been reported to have lost its capacity to transport glucose and operates as a glucose-sensitive sodium channel [135]. Facilitative glucose transporters are differentially expressed along the gastrointestinal tract [125] and generally localised to the basolateral surface to facilitate glucose efflux into the bloodstream. Transient apical translocation of glucose transporters has also been implicated in the absorption of glucose into intestinal cells [136,137]. Transient GLUT2 insertion into the brush border shortly after high glucose exposure would allow rapid glucose uptake when SGLT1 capacity is saturated and maintain the Na^+^ concentration gradient for other cellular processes, but this hypothesis remains controversial [138,139]. 

Glucose-sensitive tissues, including enteroendocrine K- and L-cells, express glucokinase (GCK), usually the rate-limiting enzyme in the breakdown of glucose [14,17,104,106,140,141]. The relatively low affinity of GCK for glucose links glycolytic fluxes to physiologically relevant extracellular glucose concentrations. However, patients with inactivating mutations in GCK did not exhibit impaired GLP-1 or GIP secretion following an oral glucose challenge [142], suggesting GCK is not the primary glucose sensor for the gut. Potassium channels sensitive to ATP (K_ATP_ channels) are described in a number of glucose-responsive tissues as a mechanism to link the nutrient status of a cell to membrane electrical excitability. The generation of ATP following catabolism of glucose increases the intracellular ATP to ADP ratio, leading to the closure of K_ATP_ channels and depolarisation of the membrane potential that may trigger action potential firing or activation of voltage-gated calcium channels to facilitate Ca^2+^ entry into the cell [143]. K_ATP_ channels are composed of a pore forming subunit (Kir6.1/2) and a sulphonylurea receptor (SUR1/2), and both Kir6.2 and SUR1 are highly expressed in GIP- and GLP-1-expressing EECs [104,106,144]. In electrophysiological recordings from GLP-1 expressing cells, glucose depolarises the cell membrane potential, triggers an increase in electrical activity and stimulates GLP-1 secretion [104,110,145]. Closure of K_ATP_ channels by tolbutamide stimulated a similar increase in action potential firing frequency and diazoxide, a K_ATP_ channel opener, blocked the effects of high glucose on cell excitability and GLP-1 release [104,145,146]. Much less is known about the electrical activity of GIP-expressing K-cells, but tolbutamide stimulated an increase in GIP secretion in primary small intestinal cultures [106] and in STC-1 cells high glucose triggered membrane depolarisation and extracellular Ca^2+^ entry to stimulate CCK release [147]. However, the finding of functional K_ATP_ channels in EECs does not imply that they are involved in glucose-triggered hormone release, and evidence that K_ATP_ channels regulate glucose-stimulated incretin hormone secretion in vivo is lacking. Mice deficient in Kir6.2 expression maintained elevated GIP release after an oral glucose load [148] and sulfonylureas had no effect on peak GLP-1 and GIP responses to an oral glucose tolerance test in human participants [149,150]. This discrepancy around the impact of K_ATP_ channels on regulating gut hormone secretion could be a result of low resting K_ATP_ channel activity in vivo compared to in vitro culturing conditions.

Other carbohydrates, such as galactose, are handled by the intestine in a similar way to glucose. However, some other dietary carbohydrates, such as fructose, can recruit different mechanisms in EECs. Fructose is not as satiating as glucose in rats [24,26] and monkeys [151] but was found to have similar or greater satiating potential in humans [152,153]. Fructose can stimulate GLP-1 and GIP secretion [60,106,115,128]; though some studies found fructose did not stimulate GIP [60,132] or GLP-1 [154] release. Fructose is not a substrate for SGLT1 but is transported into cells by the apical transporter GLUT5 [155] and possibly GLUT2. GLUT5 is abundant in the gut epithelium, with higher expression in the proximal small intestine [156]. GIP- and GLP-1 expressing cells of the intestine expressed *Slc2a5*, the gene encoding GLUT5 [104,106]. Fructose is fully metabolised, like glucose, and can recruit metabolic pathways such as K_ATP_ channel closure to induce gut hormone secretion. In GLUTag cells, fructose triggered membrane depolarisation and action potential firing, and a decrease in conductance consistent with closure of K_ATP_ channels [128]. Furthermore, fructose stimulated GLP-1 secretion was abolished by treatment with the K_ATP_ channel opener diazoxide [60], but other studies have found that mice lacking the pore-forming subunit Kir6.2 maintained fructose-stimulated GIP and GLP-1 release [157]. 

Sweet taste receptors (STRs) expressed in the gut can also detect glucose and other natural or synthetic sweeteners; however, a number of conflicting reports in the field have prevented a consensus view on their glucose-sensing role in EECs. Sweet taste receptors were first described in oral lingual taste cells [158] and comprise heterodimeric GPCRs from the type 1 taste receptor family, T1R2 and T1R3 [159]. STRs in the tongue couple to α-gustducin and activate the calcium-sensitive transient receptor potential cation channel subfamily M member 5 (TrpM5) to allow Na^+^ entry into cells and trigger membrane depolarisation. Components of the STR signalling pathway have been identified in isolated cells of the intestinal epithelium and co-localised with GLP-1 and GIP [160,161,162,163]; however, it is unclear if all elements are expressed in the same EEC to reconstitute functional STR signalling [164]. Furthermore, expression of the genes encoding the STR subunits *Tas1R2* and *Tas1R3* were not readily detectable in GIP or GLP-1-expressing cells [104,106,110]. Agonists of STRs, such as sucralose, stimulated GLP-1 release from GLUTag [165] and NCI-H716 cells [162], but did not enhance GIP or GLP-1 secretion from murine primary intestinal epithelial cultures [104,106]. Artificial sweeteners were also unable to replicate glucose-stimulated GIP [163] or GLP-1 levels [166] in rats and humans, and although mice lacking α-gustducin or T1R3 exhibited reduced GIP and GLP-1 responses [161,162], inhibiting STRs with gurmarin or blocking TrpM5 channels did not inhibit luminal glucose stimulated GIP and GLP-1 secretion [167]. Further work is needed to clarify the role, if any, of STRs, and the accompanying transduction pathway in gut endocrine cells. 

### 4.2. Proteins

Dietary proteins are broken down to small peptides by enzymes in the stomach and pancreas, then further digested by peptidases on the brush border to smaller di- and tripeptides and free amino acids. Amino acids are primarily recycled to produce proteins in the body but can also be used to generate energy when carbohydrate or lipid stores are depleted. In comparison to carbohydrates, the number of different products that arise from the breakdown of ingested proteins is vast; however, specific amino acids or small peptides can elicit specific responses from EECs. In classical studies, intraduodenal infusion of specific amino acids was more potent at stimulating GIP over CCK secretion [168]. The specificity of responses could permit targeting of peptide/amino acid-sensing pathways in incretin hormone releasing cells to manage food intake and satiety. 

Several combinations of peptides or protein hydrolysates have been shown to stimulate hormone secretion from EECs through a variety of mechanisms. Peptones, consisting of a soluble mix of amino acids and peptides derived from partial protein hydrolysis, potently stimulated GLP-1 and GIP secretion from GLUTag and STC-1 cells [100,169], which involved upregulation of proglucagon gene expression but not alteration in K_ATP_ channel activity; arguing against metabolic production of ATP from dietary protein as the primary mechanism for peptide-stimulated gut hormone secretion. In NCI-H716 cells, meat hydrolysates triggered GLP-1 release by recruiting the MAPK signalling pathway but did not alter proglucagon gene expression [170], perhaps highlighting species variation in protein/peptide-sensing in the gut. 

In murine primary colonic L-cells, the non-metabolisable dipeptide glycine-sarcosine (Gly-Sar) stimulated GLP-1 secretion [171], likely involving the H^+^/peptide co-transporter, PEPT1, as higher pH, an inhibitor of PEPT1 (4-aminomethylbenzoic acid) and knock-out of *Pept1* abolished dipeptide stimulated GLP-1 release and intracellular Ca^2+^ responses in vitro (Figure 3). Transport by PEPT1 is electrogenic and activation of PEPT1 has been shown to depolarise the cell membrane potential and activate voltage-gated calcium channels, which can contribute to hormone secretion from EECs [172]. Further work is required, however, to ascertain the role of PEPT1 as a dipeptide sensor in EECs in vivo. Similarly, the role of another electrogenic amino-acid transporter located at the apical site of the intestinal epithelium, B(0)AT-1 (*Slc6a19*), needs to be further explored. B(0)AT-1 knock-out mice showed elevated GIP and GLP-1 responses upon refeeding compared to their wild type littermates, which at least for GLP-1 might reflect increased delivery of nutrients to the more distal intestine [173].

The calcium sensing receptor, CaSR, is expressed in L-cells of the small intestine and colon and also contributes to the stimulatory responses to amino acids, dipeptides and peptones in L-cells [171,174]. Agonists of CaSR triggered GLP-1 release and selective CaSR antagonists reduced peptone-triggered GLP-1 release from rodent primary intestinal cultures [171,175]. This role of CaSR in amino acid-sensing in L-cells is comparable to its involvement in peptone triggered release of CCK from STC-1 and primary CCK-releasing EECs [176] supporting a common mechanism of amino acid sensing across EEC populations. CaSR responds to a broad range of amino acids and in other intestinal epithelial cell models the receptor couples to the phosphatidylinositol pathway resulting in elevation of intracellular Ca^2+^ [177]. In murine primary L-cells, however, peptone triggered GLP-1 release was inhibited by nifedipine and lanthanum, suggesting roles for L-type Ca^2+^ channels and TRP channels, respectively, as alternative signalling pathways downstream of CaSR recruitment [174]. In the perfused rat small intestinal model, CaSR agonists were shown to access basolaterally located receptors to trigger GLP-1 release [178]. 

There are several other GPCRs that have been implicated in amino acid responses in EECs. The lysophosphatidic acid receptor 5 (LPAR5, also known as GPR92/93), is highly expressed in the intestinal mucosal layer particularly in the duodenum [179]. This promiscuous receptor couples to various G-proteins, including Gαq and Gα12/13 [180]. LPAR5 is proposed to mediate peptone-sensing in CCK-expressing EECs [179]; however, expression of *Lpar5* transcript was not detected in colonic L-cells and GLP-1 secretion was not impaired in primary cultures from *Lpar5*-deficient mice [171] suggesting LPAR5 is not a major contributor to amino acid-sensing in L-cells. 

GPR142 is a Gαq-coupled receptor activated by aromatic amino acids such as tryptophan and phenylalanine. In vivo, oral dosing of aromatic amino acids increased plasma GIP levels which were abolished in *Gpr142* knock-out animals [181,182]. Interestingly, aromatic amino acids also evoked a rapid rise in plasma GLP-1 levels after oral gavage but this was maintained in *Gpr142*-deficient mice [182]. GPR142 seems also not to be required for gut hormone responses to dietary protein, as mixed protein delivered orally to mice triggered a robust increase in GIP and GLP-1 secretion that was maintained in *Gpr142*-deficient mice [181]. 

Taste receptors composed of T1R1/T1R3 subunits, which form the umami taste receptor, respond to a broad spectrum of aliphatic amino acids including L-glutamate [159,183]. This receptor is allosterically modulated by purine nucleotides, such as inosine-5-monophosphate (IMP), to potentiate amino acid responses. However, even though genes encoding the components of the umami taste receptor, *Tas1R1* and *Tas1R3*, were not enriched in GLP-1 or GIP-expressing cells [104,106,110], umami receptor dependent GLP-2 secretion has been reported [184]. In lingual taste cells, residual sensitivity to oral glutamate and IMP was observed in double *Tas1R1/Tas1R3* knock-outs [185] suggesting other receptors may be involved in mediating glutamate-triggered taste responses. Alternative candidate receptors including ionotropic and metabotropic glutamate receptors (mGluR1/4) have been identified in taste cells [186,187]; even though expression and a defensive role of mGluR4 activation in the duodenal mucosa has been demonstrated [188], the question of whether similar mechanisms are utilised in EECs remains to be studied.

GPRC6A is another receptor that responds to a broad spectrum of amino acids, particularly basic ones [189]. GPRC6A is activated by multiple ligands besides amino acids, including the hormones osteocalcin and testosterone, and is allosterically modulated by Ca^2+^ [190]. GPRC6A colocalised with GLP-1 expressing cells in the small intestine [191], though the majority of GPRC6A-positive cells were not immunoreactive for GLP-1. However, in another study, *Gprc6a* expression was low in primary L-cells [171] and of all the GPCRs associated with amino acid detection, *Gprc6a* was found at lowest abundance in STC-1 cells [192]. Although GPRC6A contributed to ornithine triggered GLP-1 release from GLUTag cells, ornithine did not stimulate GLP-1 secretion from primary L-cells [193]. 

More work is needed in the area of amino acid sensing mechanisms in the gut as several amino acids exert potent effects on gastrointestinal function and feeding behaviour, but the exact mechanisms remain to be fully elucidated. Tryptophan, for instance, can potently suppress energy intake and inhibit gastric emptying [194]. Although GLP-1 release from EECs can reduce food intake and slow gastric emptying rates, only modest changes in GLP-1 levels were reported by several groups following tryptophan administration [195,196] and studies that reported an increase in GLP-1 levels following tryptophan administration found GPR142 not to be involved [182]. GIP secretion was markedly increased after an oral tryptophan ingestion [181], but GIP is proposed to have the opposite effect and increase the gastric emptying rate [197]. Another amino acid with an incompletely characterised mechanism of action is glutamine, which triggers incretin hormone secretion from GLUTag, primary L- and K-cells and in human participants [104,106,198,199,200]. A sodium-dependent electrogenic transporter may be involved in glutamine sensing as both glutamine and asparagine triggered depolarising currents when applied to GLUTag cells [198] and transcripts for several electrogenic amino acid transporters are expressed in GLUTag cells. However, under non-electrogenic conditions where the cell membrane potential is pre-emptively depolarised to negate the effect of small currents generated by electrogenic transporters, glutamine could still enhance GLP-1 release [198]. Intracellular calcium and cAMP levels are elevated in GLP-1 expressing cells following glutamine application [201] suggesting the activation of additional pathways. The rise in intracellular calcium was dependent on extracellular Na^+^ and Ca^2+^, so recruitment of voltage-gated calcium channels following activation of a sodium-dependent electrogenic transporter is possible. CaSR may also be involved as the CaSR inhibitor Calhex 231 reduced glutamine stimulated GIP and GLP-1 release [174,175]. Studies in other endocrine cell models have demonstrated that CaSR can also couple to Gαs to stimulate adenylyl cyclase activity and cAMP production [202]. 

### 4.3. Fats

Fats are highly effective at suppressing energy intake and appetite [203,204], stimulating GIP secretion [118], and eliciting sustained GLP-1 release [118,205]. The stimulatory effects of dietary lipids on incretin hormone secretion were found to be dependent on fatty acid chain length and saturation [206,207,208,209]; however, other studies demonstrated the importance of metabolism and absorption for fat-stimulated GLP-1 and GIP secretion [210].

The processing of lipids after ingestion is more involved than carbohydrates or proteins. Dietary lipids in the form of triglycerides, cholesterol, phospholipids and fat-soluble vitamins are first emulsified by bile salts to promote efficient hydrolysis and absorption. Triglycerides are broken down to monoglycerides and fatty acids by pancreatic lipases released into the intestinal lumen. Hydrolysis of triglycerides is essential for EEC lipid-sensing as orlistat, a lipase inhibitor, attenuated postprandial GIP and GLP-1 secretion [211,212,213,214]. The breakdown products generated by lipases aggregate with bile acids to form micelles, facilitating uptake by enterocytes. Within enterocytes, triglycerides are re-synthesised from fatty acids and monoglycerides, and packaged with lipoproteins and other lipids to form chylomicrons. Chylomicrons are released from enterocytes from the basolateral surface and enter the lymphatic system through the central lacteal of the villus. Physiological concentrations of chylomicrons [215] significantly stimulated GLP-1 and GIP secretion in murine and human duodenal cultures [216]. The formation of chylomicrons seems to be required for lipid-stimulated release of incretin hormones as pluronic L-81, a surfactant that inhibits chylomicron synthesis [217], impaired lipid-stimulated release of GLP-1 and GIP [218,219]. Furthermore, murine knock-outs of genes responsible for re-esterification of absorbed free-fatty acids and monoglycerides prior to their incorporation into chylomicrons, monocaylglyceride-acyltransferase-2 (MGAT2) and diacylglyceride-acyltransferase-1 (DGAT1), reduced secretion of GIP from the upper GI tract following an oral triglyceride load [220]. However, MGAT2 and DGAT1 knock-out animals exhibited delayed GLP-1 responses to an oral triglyceride load, likely because impairment of fat absorption increased delivery of lipids more distally to regions of the intestine with a higher density of GLP-1-producing L-cells, leading to activation of other non-chylomicron-mediated mechanisms of GLP-1 release [221]. A neuronally mediated circuit has also been implicated in regulating feeding responses following lipid ingestion, as vagal deafferentation blocked the ability of oleate and corn oil infusions to reduce food intake [222] and silencing neuronal activity with tetracaine or capsaicin prevented duodenal lipid suppression of feeding in sham-fed rats [223,224].

The cellular mechanisms underlying lipid-stimulated GLP-1 and GIP secretion involve G-protein coupled receptors of the free-fatty acid (FFA) receptor family (Figure 4). The free-fatty acid receptor 1 (FFA1), known previously as the orphan receptor GPR40 [225,226], is highly expressed in K- and L-cells [104,106,110] and FFA1 colocalised with gut hormones GIP and GLP-1 [227]. FFA1 binds medium to long chain fatty acids and mediates lipid-induced incretin hormone release, as mice lacking *Ffar1* expression exhibited impaired release of GIP and GLP-1 following consumption of a high fat diet [227]. FFA1 couples primarily to Gαq proteins, which recruit phosphatidylinositol signalling pathways, but with some ligands has also been suggested to couple to Gαs proteins to elevate cAMP levels [228]. Heterologous expression studies suggested that Gαi/o proteins may also contribute to the downstream signalling associated with FFA1 activation [229]. In pancreatic β cells, activation of FFA1 was associated with activation of transient receptor potential cation channel subfamily C member 3, TrpC3 [230] and in mouse GLP-1 expressing cells TrpC3 activation downstream of FFA1 generated depolarising currents to increase electrical excitability and GLP-1 secretion [108]. A similar increase in electrical excitability following FFA1 activation was observed in human GLP-1 expressing L-cells, but the mechanism did not appear to involve TrpC3 channels [110]. The other known GPCR of the free-fatty acid receptor family responsive to long-chain fatty acids is FFA4, formerly known as GPR120 [231]. FFA4 has been identified in the intestine, particularly in GIP-expressing K-cells [106,232,233] and GLP-1-expressing L-cells [104,110]. FFA4 is activated by unsaturated long-chain fatty acids such as α-linoleate, docosahexaenoic acid (DHA), palmitoleate and oleate. Activation of FFA4 by long-chain fatty acids promotes secretion of GLP-1 [231] and knock-down of *Ffar4* in STC-1 cells reduced GLP-1 secretion induced by α-linolenic acid [231]. Mice deficient in *Ffar4* or treated with a pharmacological inhibitor of FFA4 exhibited reduced GIP responses to an oral lard oil challenge [233]. However, the relative contributions of FFA1 and FFA4 in mediating lipid-stimulated GIP secretion remain incompletely resolved. One study found that knock-out of *Ffar1* but not *Ffar4* reduced GIP release following an oral olive oil challenge, although double knock-out of both receptors was more effective at reducing lipid-stimulated GIP secretion than either single knock-out [234]. By contrast, another study reported that mice deficient in *Ffar1* or *Ffar4* exhibited reduced GIP secretion following oral corn oil ingestion, with a greater reduction in *Ffar1* knock-out animals [235]. In this study, it was noted that the simultaneous reduction in CCK secretion further impaired GIP secretion, likely downstream of reduced gallbladder contraction, as GIP levels were partially restored following exogenous CCK replacement, particularly in the *Ffar4* knock-out group. Like FFA1, FFA4 couples to Gαq-dependent pathways to mediate GLP-1 secretion, but Gαi/o pathways have been implicated in other FFA4 expressing cells including gastric ghrelin-secreting cells [236] and pancreatic δ cells [237]. Recruitment of protein kinase C ζ, which is not a target of diacylglycerol downstream of FFA1/4 activation, was implicated in oleic acid-induced GLP-1 release from GLUTag cells [238], although details of the pathway remain uncertain. Recruitment of the transient receptor potential channel TrpM5 has been linked to FFA4 activation by linoleic acid to stimulate CCK release from STC-1 cells [239], mirroring the involvement of TrpC3 in L-cells downstream of FFA1 activation [108]. Although the reported recruitment of different Trp channels downstream of FFA1 and FFA4 might reflect functional differences between these receptors, the possibility of cell line or species-specific responses cannot be ruled out.

Another GPCR involved in lipid sensing in the gastrointestinal tract is GPR119, which is also enriched in K- and L-cells [104,106,110,234,240]. A number of endogenous lipid substrates bind GPR119 receptors including: oleoylethanolamide (OEA), a lipid amide synthesized in the small intestine during absorption of dietary fats [241]; 2-oleoylglycerol (2-OG) and other 2-monoacylglycerols, natural digestion products of intestinal triacylglycerol digestion [242]; N-oleoyldopamine [243] and lysophosphatidylcholine [244]. The selectivity of the above-mentioned lipid-derivatives for GPR119 still needs to be addressed as OEA was able to suppress food intake in *Gpr119*-deficient mice suggesting other targets of OEA [245]. It has been reported, for example, that OEA is an endogenous substrate for the nuclear receptor PPAR-α which can also influence satiety and lipid uptake [241,246,247]. Application of OEA to various in vitro L-cell models, however, triggered GPR119-dependent GLP-1 secretion [240]. Oral administration of OEA to humans increased plasma GLP-1 and GIP levels [242], and small synthetic agonists of GPR119 elevated GIP and GLP-1 levels in mice [248] and decreased food intake in rats [249]. Whilst it is possible that local production of OEA in the small intestine modulates GPR119 activity [250], it is unclear if sufficient OEA is generated to activate the receptor [251]. Nevertheless, GPR119 seems to be an important sensor of ingested fat, as there was reduced GIP secretion following an oral triglyceride challenge in *Gpr119* knock-out animals, [234], and impaired oil-triggered GLP-1 release in mice lacking *Gpr119* in L-cells [252]. The GLP-1 response to GPR119 agonism is more prominent in the distal gut, as it was impaired in mice lacking L-cells in the terminal ileum and large intestine [28]. Mirroring this finding, GPR119 agonists were more effective at triggering GLP-1 release in vitro from murine colonic than small intestinal cultures, and elevated cAMP in ∼70% of colonic L-cells but only 50% of small intestinal L-cells [252]. By contrast, GPR119 agonism was a relatively poor stimulus of GLP-1 secretion from human colonic cultures [105], although this might reflect species-specific differences in the receptor or in vitro preparation. In a clinical study, multiple dosing of a GPR119 agonist JNJ-38431055 in subjects with T2D increased plasma GLP-1 and GIP levels, but this did not translate into improvements in 24 h blood glucose control [253]. GPR119 couples to Gαs proteins to increase cAMP levels [234,252]. Thus, in conjunction with Gαq-signalling pathways downstream of FFA1/4, the magnitude of gut hormone secretion following fat ingestion is amplified beyond activation of a single GPCR. This feature may underlie the effectiveness of fats in stimulating robust incretin hormone responses. 

As mentioned above, lipid absorption is critical for lipid-induced GIP secretion, and impairment of luminal fat digestion or epithelial absorption reduced GIP release. Receptors for free fatty acids (FFA1) are located on the basolateral face of EECs [254] so free-fatty acids can only access receptors to trigger hormone release following their absorption across the epithelium. GPR119, by contrast, is reported to be located apically as well as basolaterally [255].

Free fatty acids can enter enterocytes by the fatty acid transporter CD36. Expression of CD36 is described in oral taste bud cells [256,257] and intestine, though levels are higher in the proximal than distal intestine [258]. CD36 is an integral membrane protein which mediates the cellular uptake of long-chain fatty acids and has been linked to lingual and small intestinal fat detection and transport [241,259,260]. Mice lacking CD36 exhibit reduced chylomicron formation [261] and reduced fatty acid and cholesterol uptake in the proximal, but not distal intestine [262]. CD36 knock-out mice are also insensitive to the suppression of food intake mediated by duodenal lipid infusion [241]. Such participation of CD36 in lipid signalling mechanisms is likely indirect. As discussed above, transepithelial transport of fatty acids is necessary to target basolaterally located free-fatty acid receptors, and CD36 was also postulated to assist in FFA4 signalling by accumulating long-chain fatty acids in the vicinity of low affinity FFA4 receptors [263]. In addition, uptake of fatty acids is required for the synthesis of OEA and its precursor N-oleoyl-phosphatidylethanolamine (NOPE), and mice lacking CD36 had reduced OEA production and OEA-induced satiety [241].

### 4.4. Other: Bile Acids

Bile acids are released into the gut lumen following fat detection in the duodenum and release of CCK from EECs which stimulates gall bladder contraction. Bile acids contribute to the emulsification of dietary lipids, aiding digestion, and can also activate selective G-protein coupled bile acid receptors (GPBAR1) also known as Takeda G-protein coupled receptor 5 (TGR5) [264,265], as well as the nuclear farnesoid X receptor, FXR [266]. 

Early studies in anesthetised dogs found that intra-ileal infusion of bile increased secretion of proglucagon gene products [267,268]. EEC cell line models and mouse primary intestinal epithelial cultures confirmed that bile acids stimulated GLP-1 release [269,270,271] alongside clinical studies showing that luminal bile acids increased GLP-1 and PYY release [272,273]. Intraluminal infusion of bile into human participants and rodents also triggered a strong GIP secretory response [274,275].

Bile acid triggered gut hormone release is dependent on GPBAR1 as selective agonists of the receptor, designed with preferential activity for GPBAR1 over the nuclear bile acid receptor FXR, triggered GLP-1 release and mice lacking the receptor exhibited abrogated GLP-1 secretory responses to bile acids [112,269,276]. Ex vivo intestinal models demonstrated that only basolaterally accessible selective GPBAR1 agonists were able to elicit GLP-1 release [112,277], indicating that absorption of bile acids is critical for their effectiveness in triggering gut hormone secretion. Conjugated bile acids require transport into cells by the apical sodium-dependent bile acid transporter, ASBT [278], which is predominantly expressed in the terminal ileum. Consequently, blocking the absorption of bile acids with an inhibitor of ASBT blocked the effectiveness of bile acids to trigger GLP-1 release [112,277]. In the colon where expression of ASTB is lower, bile acids gain access to basolaterally located GPBAR1 following deconjugation and dehydroxylation by gut bacteria, which improves bile acid permeability and potency at GPBAR1 [265] and negates the dependence of bile acid-triggered GLP-1 release on ASBT [277]. Binding to GPBAR1 activates adenylyl cyclase cAMP production [112,270,271] downstream of Gαs proteins, and elevations in intracellular Ca^2+^ have also been reported in cell models of GLP-1 releasing cells following application of bile acids [112,269]. The activation of GPBAR1 in GLP-1 expressing cells increased evoked action potential activity and increased calcium currents through L-type voltage-gated calcium channels [108]. Bile acids may also increase the size of the L-cell population in the intestinal epithelium [276]. By contrast, FXR receptor activity inhibited GLP-1 production in L-cells [279], and the consequences of simultaneous activation of intestinal GPBAR1 and FXR receptors by bile acids still need to be reconciled. Bile acids may also contribute to reducing food intake by GLP-1 independent mechanisms: for example, GPBAR1 expression was found on inhibitory motor neurons in the myenteric plexus, corresponding with reduced spontaneous intestinal contractile activity following bile acid application [280]. 

### 4.5. Other: Short-Chain Fatty Acids (SCFAs)

Colonic fermentation of undigested dietary fibre by gut bacteria produces high concentrations of SCFAs in the intestinal lumen. In humans, SCFAs were most abundant in the caecum and ascending colon, whereas levels of circulating SCFA were substantially lower [27]. 

SCFA signalling is mediated by G-protein coupled receptors of the free-fatty acid receptor family. Free-fatty acid receptor 2 (FFA2), formerly known as GPR43 [281,282], is activated by SCFAs of 2–4 carbon lengths but acetate (C2) and propionate (C3) are the most potent. *Ffar2* expression is highest in the distal gut and using an antibody raised against rat FFA2, the receptor co-localised with PYY, and presumably GLP-1, in rats [283] and humans [284]. FFA2 was also described in serotonin-containing mast cells but not enterochromaffin cells, clarifying a mechanism for SCFA stimulation of colonic motility [285]. An *Ffar2*-reporter mouse line confirmed *Ffar2* expression in intestinal mast cells; however, there was weak association with EECs [286]. FFA2 is postulated to couple to G-proteins of the Gαq/11 and Gαi/o family, though the pathways recruited in EECs are incompletely elucidated. Activation of FFA2 produces a rise in intracellular calcium in murine L-cells [286,287], presumably by canonical Gαq-coupled pathways involving phospholipase C-dependent production of inositol-3-phosphate and Ca^2+^ release from intracellular stores. The free-fatty acid receptor 3 (FFA3), formerly known as GPR41 [288], is the other member of the FFA receptor family responsive to SCFAs. It preferentially binds SCFAs of three to five carbon lengths and couples to G-proteins of the Gαi/o family. FFA3 is localised to the colonic mucosa and co-localises with PYY [289]. In an *Ffar3*-reporter mouse model, a widespread pattern of *Ffar3* expression was described in EECs, including GLP-1 and GIP-expressing cells, as well as peripheral neuronal ganglia associated with enteric, sensory and autonomic signalling [286,290]. The cellular mechanism of FFA3 signalling in EECs has not been elucidated, though in neurons has been shown to involve inhibition of voltage-gated calcium channels by a Gβγ-mediated mechanism [291]. Other receptors responsive to SCFAs include the olfactory receptor subfamily 51E (OR51E1 in human, Olfr558 in mouse; OR51E2 in human, Olfr78 in mouse), and GPR109A, also known as the niacin receptor.

In murine primary colonic cultures, SCFAs stimulated GLP-1 release through FFA2-dependent [292] or both FFA2 and FFA3-dependent mechanisms [287]. A selective FFA2 agonist, Compound 1, also stimulated GLP-1 secretion [293]. SCFA-stimulated GLP-1 release did not involve pertussis toxin-sensitive pathways and SCFAs triggered a rise in intracellular calcium in GLP-1-producing EECs, suggesting Gαq-coupled pathways are responsible for mediating the stimulatory effects of SCFAs on EECs. Furthermore, a selective inhibitor of Gαq signalling FR900359 abolished SCFA-triggered GLP-1 release and an FFA2 agonist with biased activity for Gαi-signalling, AZ1729, did not trigger GLP-1 secretion [294]. The molecular details of FFA3 involvement in stimulating GLP-1 release are still unclear, given that FFA3 seems to signal exclusively through inhibitory Gαi/o-coupled pathways. In an isolated perfused colon model, basolaterally applied SCFAs triggered release of GLP-1 which was more pronounced following enhancement of cAMP levels [295]. Given the high production of SCFA luminally, localisation of receptors on the basolateral surface may be a mechanism preventing continuous activation or saturation of SCFA receptors. However, the receptors involved in enhancing GLP-1 release following SCFA administration in this ex vivo model were not clear as selective FFA2 or FFA3 agonists and antagonists had no effect. Perhaps another SCFA receptor is involved or synergistic activation of both FFA2 and FFA3 is required for mediating the effects of SCFAs on GLP-1 expressing cells. Basolateral, rather than apical sensing of SCFA on GLP-1 secreting cells is, however, further supported by the correlation of circulating rather than faecal SCFA concentrations with GLP-1 in fasting humans [296].

As mentioned previously, FFA3 co-localised with GIP in the proximal small intestine [286]. SCFAs can arise in the proximal small intestine from fermentation by oral microbiota [297] but at much lower concentrations compared with the distal intestine, or from pathological bacterial overgrowth. Generation of SCFAs in the proximal small intestine is associated with suppression of GIP release by an FFA3-dependent mechanism [298]. 

Prolonged SCFA exposure can also alter gene expression, including genes encoding gut hormones. Proglucagon expression increased in STC-1 cells after 24 h incubation with SCFAs and in the colon of rats fed chow diets enriched with fermentable fibre [299]. In human cell line models of EECs, there was little increase in proglucagon expression following SCFA treatment, but a pronounced increase in PYY expression following butyrate treatment which was mediated by histone deacetylase (HDAC) inhibition and FFA2 [300]. SCFA have also been reported to alter the number of GLP-1 producing L-cells, but whereas an increase in L-cell number was observed following chronic SCFA exposure in mouse and human intestinal organoids in vitro [301], SCFA appeared to suppress L-cell number and function in germ free mice recolonised with SCFA producing bacteria [302,303].

## 5. Effect of GLP-1 and GIP on Food Intake and Weight Loss

The consensus view is that exogenous administration of GLP-1 reduces energy intake and appetite in rodents [304] and humans [305,306,307]. The anorexigenic actions of GLP-1 extend to long-acting mimetics such as exenatide, liraglutide and semaglutide [308,309,310,311] and persist in Type 2 diabetic and obese individuals, thus supporting the use of GLP-1 receptor agonists in weight management therapies.

The effect of GIP on food intake is more controversial. Administration of GIP or a long-acting analogue acyl-GIP, at doses sufficient to produce positive metabolic effects, had no effect on body weight in mice [312] or hunger scores and food ingestion in humans [197], and in a recent double-blind crossover study, there was no significant change in food intake in overweight/obese individuals given GIP along with an IV glucose infusion that mimicked the blood glucose rise after an oral glucose challenge [313]. Animal models with diminished GIP activity including *Gipr* knockout mice [314], enteroendocrine K-cell ablation [315], and GIPR antagonism [316,317] show reduced body weight gain associated with diet-induced obesity. By contrast, overexpression of GIP in mice also led to decreased energy intake and reduced weight gain associated with diet-induced obesity [318]. Interpretation of these studies may be complicated by confounding factors such as GIP stimulation of pancreatic insulin and glucagon secretion, which can themselves modulate appetite [319,320], and compensatory changes in the absence of functional GIP signalling [321]. Furthermore, GIP has a complicated pharmacology as certain peptides declared to be antagonists of GIPR in fact display partial agonist activity [322,323] and interspecies differences in GIPR pharmacology exist [322]. Recent studies have favoured a reduction of food intake as a consequence of pharmacological GIPR agonism [324], but further studies will be required to fully elucidate the molecular mechanisms governing GIP action on food intake. 

Despite the controversy surrounding GIP action on food intake, there is growing interest in developing agonists that target two or more receptors involved in metabolic control, including receptors for the incretin hormones GLP-1 and GIP. Studies in mice demonstrated that co-administration of GIP and GLP-1 analogues enhanced body weight loss and improved glycaemic control in obese mice [312,325,326]. Dual GIP/GLP-1 agonists are in development as potential new anti-diabetic and anti-obesity treatments. The pharmacological reasoning for this approach is to amplify the beneficial metabolic effects whilst lowering doses and adverse side effect profiles of the individual components. Promising improvements in blood glucose control were obtained with the acylated unimolecular dual GIP/GLP-1 agonist RG7697/NNC0090-2746 in healthy [327] and T2D patients [328] and significant reductions in body weight were achieved but the magnitude was possibly similar to liraglutide treatment alone [329]. However, another dual GIP/GLP-1 agonist tirzepatide (LY3298176) displayed improved efficacy over the selective GLP-1 receptor agonist dulaglutide [330,331]. 

## 6. Gut Hormone-Mediated Mechanisms of Satiety

Infusion of various nutrients into the small intestine is associated with greater suppression of food intake than nutrients that are delivered intravenously [332,333], strongly supporting a gut-derived mechanism for satiety. GLP-1 and GIP released from the gastrointestinal tract can initiate satiety signals through a variety of mechanisms ranging from direct modulation of gastrointestinal function to recruitment of central circuits involved in feeding behaviours.

The actions of GLP-1 and GIP on gastric emptying and gastric acid secretion are well studied. Briefly, postprandial GLP-1 is associated with delayed gastric emptying [334] whereas GIP was found to have the opposite effect of increasing the gastric emptying rate [197]. Vagal afferent fibres were necessary for mediating GLP-1 effects on gastric emptying [335] along with cholinergic signalling and GLP1R activation [336], which is consistent with GLP1R expression described in the gastric antrum and pylorus [337]. Other gut hormones such as CCK and PYY also control gastric emptying rates [338]. Vagal innervation was also necessary in mediating the inhibitory effects of GLP-1 on gastric acid secretion [339] as vagotomised patients did not exhibit an inhibitory response to GLP-1 [340]. Peripheral infusion of GIP also inhibited gastric acid secretion, albeit at supraphysiological concentrations [341] and involved a reduction in postprandial gastrin release. 

Gut hormones also internally regulate intestinal motility and luminal transit in response to a high nutrient load. Infusion of physiological levels of GLP-1 slowed intestinal motility in the fed and fasted state in rats [342] and humans [342,343], and was reversed by the GLP1R antagonist Exendin(9–39) [342]. Exogenous GIP also reduced intestinal transit in mice, which was blocked by a somatostatin receptor antagonist [344], but by comparison GLP-1 was more potent than GIP in abolishing myoelectric activity in the small bowel of rats [345].

Overall, by modulating various aspects of gastrointestinal function (gastric emptying, gastric acid secretion and intestinal motility), GLP-1 and GIP can slow the digestion of nutrients allowing more efficient nutrient breakdown and absorption. This serves to increase exposure of proximal EECs to nutrients. The slowing of gastrointestinal activity can also increase distension of the stomach and intestinal wall, and consequent activation of stretch-activated mechanoreceptors directly signals to the brain to control feeding and enhances the perception of fullness [346]. 

Neuronal mechanisms play an integral role in mediating the satiating effects of GLP-1 and possibly GIP. The effect of peripherally administered GLP-1 on food intake was ablated after bilateral subdiaphragmatic total vagotomy [25] in rats, and the suppression of short-term food intake following intraduodenal nutrient infusion was eliminated by selective vagal rhizotomy of the celiac branch, which abolishes afferent vagal inputs to the intestine [26]. Both findings highlight the significance of vagal signalling in appetite control. Vagal afferent neurons project peripherally to visceral organs, including much of the small intestine and the proximal third of the large intestine. A subpopulation of vagal afferent neurons express receptors for GLP-1 [337,347,348] but single-cell transcriptomics studies were unable to resolve expression of receptors for GIP in nodose neurons. GLP-1 directly stimulated activity in vagal afferent neurons [349] and augmented Ca^2+^ responses in GLP1R-expressing nodose neurons in vitro [337]. Vagal afferent neurons project to two brainstem regions: the area postrema and the nucleus of the solitary tract (NTS). GLP1R expression, staining and ligand binding have been detected in a number of brain regions, including the area postrema, ventromedial hypothalamus, arcuate and paraventricular nuclei [350,351,352,353]. Sensory afferent inputs into the hindbrain are essential for mediating the suppression of feeding by nutrients as chemical destruction of these neurons by capsaicin abolished this inhibition [354,355]. Some neurons in the NTS also produce and release GLP-1 [356] but these neurons do not express GLP1R or respond to exogenous GLP-1 [357,358]. Although vagal afferent signalling is important in mediating some effects of GLP-1 on feeding control, the route of intestinal GLP-1 to activate vagal afferent neurons is unclear. Vagal afferents, particularly neurons that express gut hormone receptors, do not appear to innervate the basolateral surface of L-cells [348,359], although *Glp1r*-expressing enteric neurons have been identified [337], which may be involved in relaying signals. Diffusion of GLP-1 in the vicinity of its site of release from L-cells may contribute to non-synaptic activation of local nerve endings, and could act in tandem with other mediators released from EECs such as glutamate [360] or ATP [361] to exert its effects on food intake.

Central injection of GLP-1 into the cerebral ventricles [304,362,363,364] or hypothalamus [365] reduces food intake. These centrally mediated effects of GLP-1 were dependent on GLP1R signalling as they were not observed in *Glp1r* KO animals [366] or in the presence of the GLP1R antagonist exendin(9–39) [362,367]. Whilst some of the central GLP1R sites are involved in the satiating effects of peripherally supplied GLP1R agonists, it is becoming clear that activation of GLP-1 expressing neurons in the NTS recruits additional anorexic circuits [358].

There is increasing support for a central mechanism mediating food intake reduction by GIP. Central administration of GIP decreased food intake and body weight and combined with GLP-1 produced a synergistic reduction of food consumption [367]. Furthermore, *Gipr* expression has been described in the arcuate, dorsomedial, and paraventricular nuclei in the hypothalamus and activation of *Gipr* expressing cells in this region suppressed food intake [368]. Many questions remain, including whether circulating GIP released from K-cells in the proximal small intestine activates centrally expressed GIPR, which are also found in the area postrema [368], as well as the identity and relative importance of the circuitry mediating GIP-mediated suppression of food intake. 

Besides its role in mediating homeostatic feeding, GLP-1 is also implicated in circuits driving food reward and motivation. Activation of GLP-1 producing preproglucagon neurons in the NTS seems to be of minor importance for homeostatic feeding control, but becomes relevant under stress, one of which is overeating, presumably involving stomach stretching [369]. In this context it is important to note that *Glp1r* is expressed by neurons in the ventral tegmental area (VTA) and nucleus accumbens, brain regions associated with reward and desire. Exendin-4, a GLP-1 mimetic, or GLP-1 injected peripherally reduced palatable food intake and reward-motivated behaviour and a similar effect was observed when exendin-4 was injected directly into the VTA [370,371]. In agreement with these findings, injection of the GLP1R antagonist, Exendin(9–39), into the nucleus accumbens increased meal size and palatability of sucrose solutions [372]. 

## 7. Future Directions

There are still many unresolved issues in our understanding of nutrient-induced gut hormone secretory mechanisms, including the role of GLP-1 produced in the distal GI tract. GLP-1 is produced in large quantities in the distal small intestine and colon, regions of the intestine not typically reached by ingested nutrients, where its release may follow different regulatory mechanisms including control by metabolites of gut microbiota and neurohormonal pathways. Recent studies identified the possible importance of distal L-cells in mediating responses to lipid-derivatives, melanocortin receptor 4 agonists and lipopolysaccharides [28]. Another question is whether therapeutic activation of GPCRs in EECs could trigger sufficient gut hormone release to produce significant metabolic benefits or satiation. Injectable GLP1 receptor agonists and the levels of GLP-1 achieved following bariatric surgery are far in excess of levels expected in a normal healthy postprandial state. The feasibility of targeting nutrient-sensing receptors in EECs also needs to be properly assessed. Many of these receptors respond to a range of ligands but the potential importance of biased downstream signalling is not known. Finally, desensitisation of nutrient-sensing receptors in EECs has not been investigated in-depth, but could arise from the prolonged exposure of EECs to saturating concentrations of nutrients in the postprandial state or with altered dietary status such as in models of diet-induced obesity, which have been shown to alter expression levels of peptide hormone receptors expressed on vagal afferent neurons [373]. Further complications or, alternatively considered, treatment options arise from cross talk between different EECs, notably the apparent chronic paracrine inhibition of L-cells by somatostatin [374] released from nearby D-cells, which themselves express a number of nutrient sensing receptors, as shown by RNAseq for D-cells in the stomach [375]. 

## 8. Conclusions

Nutrient absorption and stimulation of gut hormone secretion occur predominantly in the proximal small intestine. However, as demonstrated by the effectiveness of bariatric surgery to deliver nutrients more distally along the gut, distal GLP-1 producing L-cells represent a vast source of endogenous GLP-1 that can potentially be exploited to suppress appetite and increase insulin secretion. Future work is needed to reveal the physiological importance of distal GLP-1 producing L-cells and identify strategies to recruit these cell populations for improved glucose homeostasis and appetite regulation.

## Figures and Tables

**Figure 1 nutrients-13-00883-f001:**
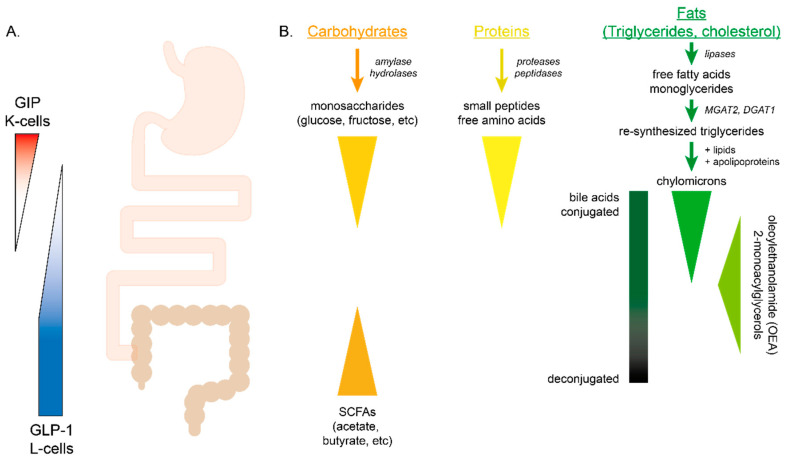
Overview of macronutrient digestion. (**A**) Schematic representation of the distribution of glucose-dependent insulinotropic polypeptide (GIP) expressing K-cells and glucagon-like peptide 1 (GLP-1) expressing L-cells along the longitudinal intestinal axis. (**B**) Schematic representation of major sites of macronutrient digestion and absorption for carbohydrates, proteins and fats along the longitudinal intestinal axis. The breakdown of macronutrients denoted above primary location of nutrient absorption. Whereas absorbed monosaccharides and amino acids are exported from the intestinal epithelium as such, the majority of free fatty acids and monoglycerides are stepwise re-synthesised within the epithelium by MGAT2 and DGAT1 into triglycerides, which together with other lipophilic substances are secreted as chylomicrons. The production of lipid metabolites, such as OEA and 2-monoacylglycerides, which are synthesized following absorption of dietary fats, is represented in light green. Conjugated bile acids released following fat detection in the proximal small intestine is deconjugated in the distal intestine by colonic gut bacteria, as indicated by a dark green bar. Few macronutrients escape absorption in the small intestine, but bacterial fermentation of “indigestible fibres” provides SCFA as another nutritional source in the large intestine. Abbreviations: DGAT1, diacylglyceride-acyltransferase-1; GIP, glucose-dependent insulinotropic polypeptide; GLP-1, glucagon-like peptide 1; MGAT2, monoacylglyceride-acyltransferase-2; OEA, oleoylethanolamide; SCFAs, short-chain fatty acids.

**Figure 2 nutrients-13-00883-f002:**
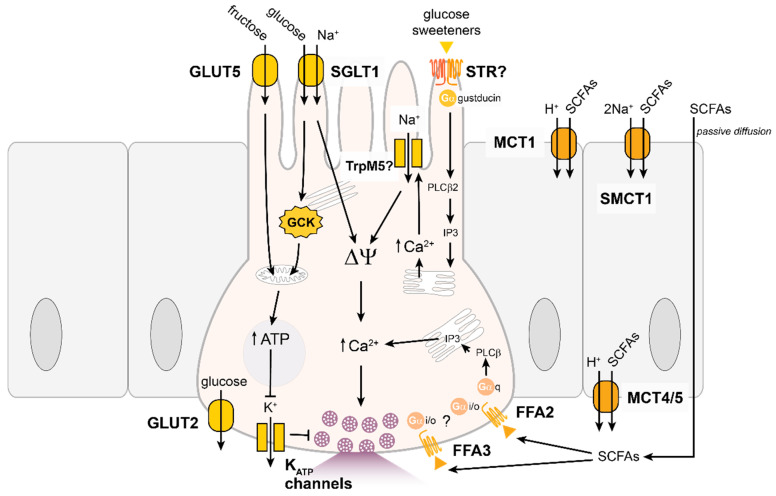
Carbohydrate-sensing mechanisms in incretin hormone secreting EECs. Schematic of EEC with apical process (top) and basolateral surface (bottom) populated with secretory vesicles containing incretin hormones. Enterocytes shown in grey beside EEC. Glucose sensing by incretin-secreting EECs is critically dependent on sodium coupled uptake via SGLT1. Other sensors have been described, but are controversial (labelled with a question mark) or appear to be of limited physiological relevance. When SGLT1 is inhibited, GLP-1 secretion is elevated at later time points, possibly downstream of fermentation to short-chain fatty acids, which target G-protein coupled fatty acid receptors FFA2 and FFA3. These have been depicted on the basolateral membrane in analogy to the location of FFA1 (see text for details). Transport mechanisms of SCFAs across the intestinal epithelium illustrated in enterocytes on the right. Abbreviations: ATP, adenosine triphosphate; FFA2 or 3, Free fatty acid receptors; GCK, glucokinase; GLUT2, glucose transporter 2; GLUT5, glucose transporter 5; IP3, inositol triphosphate; K_ATP_, ATP sensitive potassium channel; MCT1 or 4/5, monocarboxylate transporters; PLCβ, phospholipase C beta; SCFAs, short-chain fatty acids; SGLT1, sodium-glucose linked transporter 1; SMCT1, sodium-coupled monocarboxylate transporter 1; STR, sweet taste receptor; TrpM5, transient receptor potential cation channel subfamily M member 5; ΔΨ, membrane depolarization.

**Figure 3 nutrients-13-00883-f003:**
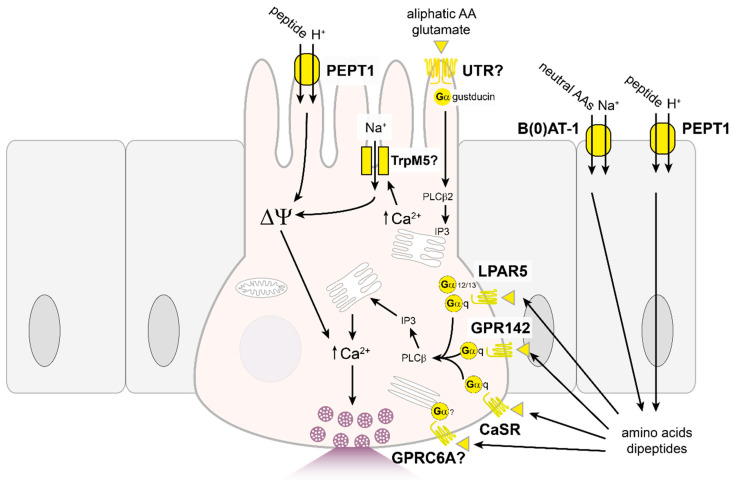
Protein-sensing mechanisms in incretin hormone secreting EECs. Schematic as in Fig2. Good evidence exists for a role of PEPT1, CaSR and GPR142 in incretin secretion, however, the exact locations of receptors and transporters are incompletely defined. Controversial contributors are marked with a question mark (see text for details). Abbreviations: AA, amino acid; B(0)AT-1, sodium-dependent neutral amino acid transporter; CaSR, calcium-sensing receptor; GPR142, G-protein coupled-receptor 142; GPRC6A, G protein-coupled receptor family C group 6 member A; IP3, inositol triphosphate; LPAR5, lysophosphatidic acid receptor 5; PEPT1, H^+^/peptide co-transporter; PLCβ, phospholipase C beta; TrpM5, transient receptor potential cation channel subfamily M member 5; UTR, umami taste receptor; ΔΨ, membrane depolarization.

**Figure 4 nutrients-13-00883-f004:**
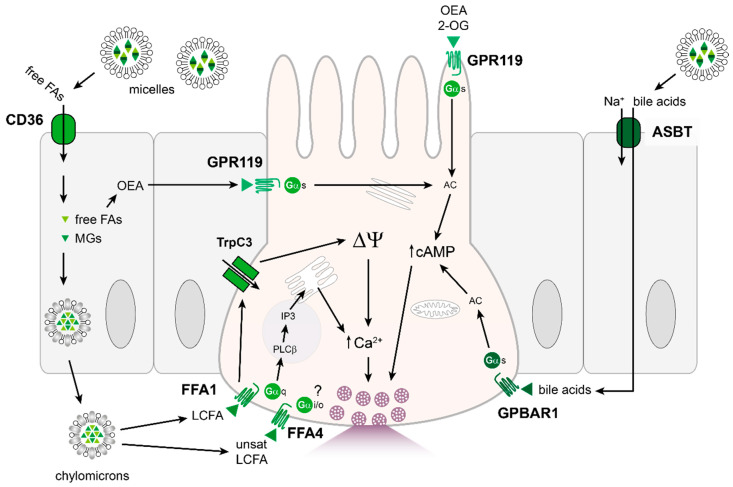
Fat-sensing mechanisms in incretin hormone secreting EECs. Lipid sensing appears dominated by activation of G-protein coupled receptors. Contrary to what had been thought originally, these do not directly sample the luminal contents, but are shielded behind lipid absorption by enterocytes, with the possible exception of the mono-acyl-glyceride sensor GPR119. Absorption of micelles and formation of chylomicrons through enterocytes is illustrated on the left. Bile acids, which assist in the emulsification and breakdown of fatty acids, also stimulate incretin hormone secretion and the mechanism of bile acid-sensing in EECs is illustrated in dark green on right side of EEC. Transport route for conjugated bile acids in the small intestine via ASBT, illustrated in enterocyte on the right. See text for details. Abbreviations: 2-OG, 2-oleoylglycerol; AC, adenylyl cyclase; ASBT, sodium-dependent bile acid transporter; cAMP, cyclic adenosine monophosphate; CD36, cluster of differentiation also known as fatty acid translocase; FA; fatty acid; FFA1, free fatty acid receptor 1; FFA4, free fatty acid receptor 4; GPBAR1, G-protein coupled bile acid receptor 1; GPR119, G-protein coupled receptor 119; IP3, inositol triphosphate; LCFA, long-chain fatty acid; MG; monoglycerides; OEA, oleoylethanolamide; PLCβ, phospholipase C beta; TrpC3, transient receptor potential cation channel subfamily C member 3; ΔΨ, membrane depolarization.

## Data Availability

Data sharing not applicable.

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
