# Peer review of "Nutrient-Induced Cellular Mechanisms of Gut Hormone Secretion"

_nutrients, 2021, doi:10.3390/nu13030883_

Round 1
Reviewer 1 Report
Lu et al have written a very thorough and comprehensive review of the nutrient-induced cellular mechanisms of gut hormone secretion, particularly of GLP-1 and GIP. The topic and content will be of interest to many researchers and this chapter provides a great resource compiling data published to date. The fat-sensing mechanisms and accompanying Figure 4 are particularly informative.
My only major critique concerns Figure 1. While the arrows indicating gradients of absorption for carbohydrates and proteins are clear and well-described in the figure legend, the panel describing absorption for fats is confusing and poorly described in the legend. What is the significance of the different shades of green? The way this figure and legend is presented suggests that enzymes MGAT2 and DGAT1 are restricted to proximal small intestine, and chylomicrons are only synthesized there. Is this accurate? Moreover, the gradient bar figure for bile acids and the gradient for OEA and 2-monoacylglycerols are not described in the legend, and thus, I do not understand the context or significance of these figures.
Minor concerns:
The authors should take care to denote which results were found via protein/hormone expression, and which results were found via transcriptional expression. This is particularly relevant for lines 110-185.
In Figure 2, the SCFAs and FFA2 and FFA3 are drawn on the basolateral side of the EEC. Are these receptors not localized on the apical side? Adding the mechanism of how SCFAs arrive at the basolateral side would be helpful (as is done in Figures 3 and 4).
In the legend for Figure 4, there is a near-identical duplication of sentences (lines 601-603).
“In vitro” should be italicized in line 635.
“Neurons” was misspelled in line 885.
It appears that typesetting of the manuscript resulted in loss of Greek characters throughout the manuscript: Lines 140, 379, 390, 455, 460, 515, 558, 559, 560, 562, 571, 574, 586, 588, 608, 619, 638, 639, 690, 716, 718, 722, 728, 736, 738, 739, 741.
Author Response
In response to Reviewer 1:
My only major critique concerns Figure 1. While the arrows indicating gradients of absorption for carbohydrates and proteins are clear and well-described in the figure legend, the panel describing absorption for fats is confusing and poorly described in the legend. What is the significance of the different shades of green? The way this figure and legend is presented suggests that enzymes MGAT2 and DGAT1 are restricted to proximal small intestine, and chylomicrons are only synthesized there. Is this accurate? Moreover, the gradient bar figure for bile acids and the gradient for OEA and 2-monoacylglycerols are not described in the legend, and thus, I do not understand the context or significance of these figures.
Although MGAT2 and DGAT1 expression is found throughout the gastrointestinal tract, absorption of dietary fats and chylomicron formation is concentrated in the small intestine, which is denoted in Figure 1. We have added in the sentence, “The breakdown of macronutrients denoted above primary location of nutrient absorption” to clarify the intention of the figure. As well, we added the following descriptions for bile acid and lipid metabolites (OEA) that were missing in the previous manuscript version: “The production of lipid metabolites, such as OEA and 2-monoacylglycerides, which are synthesized following absorption of dietary fats, is represented in light green. Conjugated bile acids released following fat detection in the proximal small intestine is deconjugated in the distal intestine by colonic gut bacteria, as indicated by a dark green bar.”
The authors should take care to denote which results were found via protein/hormone expression, and which results were found via transcriptional expression. This is particularly relevant for lines 110-185.
Thank you for this comment. We have clarified which results from co-localisation of GLP-1/GIP with other gut hormones were from protein/hormone expression studies or transcriptional expression studies (lines 115-181).
In Figure 2, the SCFAs and FFA2 and FFA3 are drawn on the basolateral side of the EEC. Are these receptors not localized on the apical side? Adding the mechanism of how SCFAs arrive at the basolateral side would be helpful (as is done in Figures 3 and 4).
Although it has not been definitively localised to the basolateral side of EECs, SCFA receptors would benefit from shielding from the high concentrations of SCFAs produced in the gut lumen, similar to receptors responsive to long-chain fatty acids (FFA1/4). As suggested by Reviewer 1, we have added in Figure 2 details of how SCFAs are transported across the intestinal epithelium which would permit access to basolaterally located receptors. We have also added an additional reference reporting correlation of fasting GLP-1 with circulating rather than faecal SCFA concentrations – of note, this report failed to find a similar correlation with fasting PYY, but nonetheless would further support a sensing of SCFA on the basolateral side of L-cells. The failure to stimulate GLP-1 secretion in the perfused rat intestine with selective FFA2/3 agonist had already been discussed – this somewhat prevents a clear basolateral localisation, as has been done for FFA1 – a sentence that the location is in analogy to FFA1 has thus additionally been added to the Figure legend.
In the legend for Figure 4, there is a near-identical duplication of sentences (lines 601-603).
The wording in Figure legend 4 was adjusted to sound less repetitive.
The following minor corrections were also made as suggested by Reviewer 1:
“In vitro” should be italicized in line 635.
“Neurons” was misspelled in line 885.
It appears that typesetting of the manuscript resulted in loss of Greek characters throughout the manuscript: Lines 140, 379, 390, 455, 460, 515, 558, 559, 560, 562, 571, 574, 586, 588, 608, 619, 638, 639, 690, 716, 718, 722, 728, 736, 738, 739, 741.
Reviewer 2 Report
This is an excellent review of the nutrient-sensing mechanisms involved in GLP-1 and PYY secretion. The review is focussed on nutrient-stimulation of GLP-1 and GIP along with the cellular mechanisms involved, and is very well referenced. The key mechanisms are summarised in Figures. The content of this manuscript goes beyond previous reviews of the subject. The discussion covers everything from cellular mechanisms through to diet and health, and will likely appeal to a broad readership. It is accurate and well written.
Author Response
Thanks for your comments.
Reviewer 3 Report
Very good and comprehenisve review on gut-hormone secretion. A new perspective on old subject
Author Response
Thanks for your comments